# How we compare: A new approach to assess aspects of the comparison process for appearance-based standards and their associations with individual differences in wellbeing and personality measures

**Peter A. McCarthy**, **Thomas Meyer, Mitja D. Back, Nexhmedin Morina**\*

Institute of Psychology, University of Münster, Münster, Germany

\* morina@uni-muenster.de

## Abstract

We introduce a novel approach to assess habitual comparison processes, while distinguishing between different types of comparison standards. Several comparison theories (e.g., social) suggest that self-evaluations use different standards to inform self-perception and are associated with wellbeing and personality. We developed the Comparison Standards Scale for Appearance (CSS-A) to examine self-reported engagement with social, temporal, criteria-based, dimensional, and counterfactual comparisons for upward and downward standards in relation to appearance. The scale was completed by three hundred participants online alongside measures of appearance schemas, social comparison evaluations, depression, anxiety, stress, self-esteem, physical self-concept, narcissism, and perfectionism. The CSS-A was found to reliably assess individual differences in upward and downward comparison frequency and affective impact for multiple comparison standards. In line with theory, CSS-A upward comparisons were more frequent than downward comparisons and coincided with negative (versus positive) affective impact. Comparison intensity (i.e., comparison frequency × discrepancy) predicted negative and positive affective impact for upward and downward comparisons, respectively. This relationship was partially mediated by appearance concern for upward comparisons (a composite of appearance schemas and physical self-concept), yet moderated by negativity for downward comparisons (a composite of depression, anxiety, stress, and self-esteem). We offer a framework for measuring the comparison process that warrants further research on underlying comparison processes, for which the CSS(-A) and experience sampling methods should serve as useful tools.

## Introduction

Comparison processes are a ubiquitous and important phenomenon of everyday life. They are essential for people's self-evaluation and can pertain to different standards against which they

**Competing interests:** The authors have declared that no competing interests exist.

compare themselves (e.g., social, temporal, counterfactual), different dimensions (e.g., body image, academic achievement, social rank), and different directions of comparison (e.g., upward, downward, lateral). Comparisons can result in different engendered affective and cognitive reactions, thereby influencing everyday behaviour. Notably, people can differ in the propensity and strength with which they habitually engage in certain types of comparison. However, despite their potential importance, these differences have not yet been systematically described in concert and there exists no overarching approach to their measurement. Using appearance-related comparisons as an example, we provide (a) a new approach to disentangle different comparison directions and standards and (b) a new scale–the *Comparison Standards Scale for Appearance (CSS-A)*–that captures individual differences therein. We also explore how individual differences in appearance-based comparison processes are associated with wellbeing and personality factors.

## Comparison standards

Comparative thinking with respect to self-perception involves judgements of a person's abilities or attributes relative to a comparison *standard*. Several theories propose that individuals make use of certain types of comparison when evaluating their abilities, attributes, and wellbeing [1–3]. Most prominently, social comparison theory [4,5] asserts that individuals will use comparisons with others to make self-evaluations on specific attributes, with their perceptions of other people serving as the standard. Similarly, temporal comparison theory [6] proposes that individuals compare with a representation of the self in the past or future, which serves as the comparison standard for evaluating the perceived current self. Dimensional comparison theory [7] also proposes an intra-individual mode of comparison, where domains of ability or attributes of the individual are used as the comparison standard. The theory posits that individuals compare their own abilities in a specific target domain (e.g., math skills) with another comparison domain (e.g., verbal skills) as part of self-evaluation.

Outside of the social, temporal, and dimensional theoretical frameworks, scholars have also described criteria-based and counterfactual comparisons. Criteria-based comparisons involve internalized expectations of ideal or feared selves, and 'ought' selves [8,9], reflecting a degree of respective excellence/inferiority or adequacy. These manifest as comparisons of the perceived current self with idealized standards (e.g., achieving the highest grades in the school, excellence), feared standards (e.g., getting the lowest grades, inferiority), or socially shared or codified rules (e.g., achieving at least the average grade, adequacy). Counterfactual comparisons involve comparing oneself to an expected or desired hypothetical self that deviates from reality [10]. Such comparisons involve thinking about alternative outcomes that serve as the standard, such as if something had (additive) or had not (subtractive) happened [11]. Taken together, various theoretical accounts place comparison processes at centre stage in the evaluations of one's self-perceptions. Social comparison theory is the most widely researched, informing most of what we know about the comparison process, whereas other comparison standards have been investigated to a much lesser degree and have largely been investigated in isolation, despite shared aspects of comparison. We argue that to better understand the role of comparisons in self-perception, types of comparison standards need to be considered collectively.

## Characteristics of the comparison process

Comparison is best defined as a process consisting of three broad stages of acquiring, evaluating, and reacting to information [1,2,12]. For example, Wood [12] identifies the social comparison process beginning with the evaluation stage, where social comparison information is

considered in relation to some attribute of the self, followed by a reaction depending on the outcome of the evaluation. The ability or attribute that is being evaluated in the process is referred to as the comparison *dimension* (e.g., appearance). Relative to the target, which represents one's own self-perception, standards can be perceived as better, similar, or worse on the given dimension, representing the *direction* of comparison, namely upward, lateral, or downward. For example, I might compare my appearance (i.e., the target) with a friend and evaluate them to be better-looking (upward social), or to myself in the past and find that I used to look worse (downward past temporal). Next to direction, a critical aspect of the comparison outcome is the degree of perceived *discrepancy* between the target and the comparison standard. In the case of my appearance, a large perceived discrepancy between the target and social upward standards translates into unfavourable evaluations of my current appearance.

## Affective consequences of comparison

Several studies have examined the affective consequences associated with comparisons, with the majority focused on social comparisons in context of appearance and wellbeing. For example, comparisons with upward social standards have often been found to be associated with a worsening of affect, whereas comparisons with downward standards are usually associated with improvements in affect, albeit less reliably [2,5]. In addition, the relationship between comparison direction and affective consequence may depend on the degree of perceived (dis) similarity with social standards. For example, positive consequences of upward social comparisons (i.e., improved affect, self-esteem, and self-evaluation) have also been found as a function of expected similarity with the standard. Arigo and colleagues [13] reviewed social comparison processes in chronic health conditions such as cancer and found that positive affective consequences were reported among individuals who identified with an upward coping standard, or who contrasted with a downward standard for illness severity. Additionally, negative affective consequences tended to be reported for individuals identifying with downward standards, such as others who are very ill or coping poorly. Thus, upward and downward comparisons can increase positive or negative affect depending on perceived similarity or discrepancy.

Other comparison standards have been investigated to a lesser degree regarding affective consequences, however they represent a growing body of research. In a rare multi-standard experience sampling study considering social, temporal, and counterfactual comparative thinking, Summerville and Roese [14] found that counterfactual and past-temporal upward comparisons were associated with more positive affective responses, whereas social and future temporal upward comparisons showed no pronounced negative or positive responses. Taken together, these findings suggest that affective consequences can vary according to the type of comparison standard and direction, and that an integrative approach to comparison assessment is necessary to better understand the role of comparisons in wellbeing. We are not aware of any research that has attempted to assess how affective consequences vary across multiple types of comparison standards, where standards and direction are specifically defined.

## Assessment of individual differences in the comparison process

To date, assessments of individual differences in the comparison process have typically focused on isolated aspects of the larger comparison process [2,15]. For example, the Social Comparison Scale (SCS) [16] is the most commonly used comparison scale, which measures self-evaluation relative to others across various dimensions in context of social rank, attractiveness and group fit. An SCS score provides an aggregate rating of social comparison outcomes where higher scores suggest more favourable self-perception scores, which allows for general between-person analyses of self-evaluation via social standard(s). The Iowa-Netherlands

Comparison Orientation Measure (INCOM) [17] is another common social comparison measure that assesses the tendency of individuals to engage in comparison-related cognitions and behaviours, which provides an aggregate rating of social comparison orientation, i.e., the likelihood of engaging and using social standards. While the INCOM and SCS are useful for comparing individual differences in inclination and general self-evaluation of social comparison, we still require more specific measures of individual differences in processes that contribute to engagement of information and the evaluative outcomes, such as who is used as the standard (s), what degree of (dis)similarity there is with standards, how often do comparisons occur, what is the comparison direction, and what are the consequences of evaluations.

A more comprehensive measure of social comparison is the Rochester social comparison diary [18], in which individuals record the frequency of daily comparisons, along with the corresponding dimensions, social standards (i.e., who), similarity, and pre- and post-affect. It was used effectively to demonstrate individual differences in college students, where the direction of the comparison varied depending on the standard used (e.g., close friend or stranger), the precomparison mood, and self-esteem. Upward comparisons were associated with decreases in subjective wellbeing, whereas downward comparisons were associated with enhancements. Similar results were also reported in social anxiety disorder patients versus healthy controls [19], and high and low dysphoria groups [20]. The comparison diary has also been adapted to dimensional comparisons [21], where a contradictory pattern to the social comparison direction and subsequent affect was found, indicating that dimensional comparisons have a different motivational significance for self-perception and that comparison processes will vary depending on the standard used [1]. Diary studies therefore show the importance of disentangling the different aspects of the comparison process. However, very few cross-sectional or experience sampling studies have attempted this beyond the field of social comparison and almost none have attempted this for multi-standard comparisons. To address this, we present the development of a dimension-based self-report measure for multiple comparison standards that assesses individual differences in aspects of the comparison process more comprehensively.

### The case for appearance-based comparisons

To investigate properties of habitual comparisons with multiple standards and their affective consequences, we decided to focus on one's perceived appearance as the comparison dimension, given its prominence as a salient construct for individuals in modern daily life, as well as its well-established link with comparison processes. For instance, the frequency of social comparison has been found to predict body dissatisfaction and appearance schemas, with frequency and evaluation factors predicting eating disorder symptoms, [e.g., 22,23]. This is reflected in the DSM-5 criteria for body dysmorphic disorder, where criterion B includes repetitive mental acts such as comparing one's appearance with that of others, i.e., social comparison [24]. While criteria-based comparisons are not named as DSM-5 criteria, discrepancies between perceived body shape and idealised body shape in females have been found to predict eating disorder pathology [25]. Other types of comparison standards can also potentially offer useful insight, despite a lack of research in the literature. For example, appearance is susceptible to temporal effects of ageing, and pathological body image disturbances may involve counterfactual beliefs (e.g., if my nose were smaller, I would be more beautiful).

### Comparison correlates of mental health and personality

Mental wellbeing and personality factors may act as mediators or moderators of the comparison process. A recent systematic review of social comparison processes suggests that mental

health status contributes to differences in social comparison habits, where clinical and subclinical levels of depression and anxiety are associated with maladaptive comparisons [2]. These involve a higher likelihood of selecting upward comparison standards, [e.g., 26], and frequently making upward comparisons across multiple dimensions [19], or comparing on specific dimensions that are congruent to depressive personality styles [20]. Meta-analyses have also found significant associations between depression symptoms and negative self-evaluations relative to others, as measured by the SCS [2,27]. Similarly, a meta-analysis of upward counterfactual thinking found significant associations with depression ratings [28], and recent findings suggest that overall counterfactual comparisons explain significant variance in posttraumatic stress symptoms [29]. Effects of depression are also seen in temporal comparisons, where (unfavourable) upward past comparisons were more likely in a depressed group, and (favourable) downward comparisons were more likely in a healthy group [30]. Considering the evidence of relationships between psychopathology and comparisons, we investigate associations of depression, anxiety, and stress with aspects of the comparison process across standards.

Perfectionism and narcissism have also been identified in the literature as potential moderators of the comparison process, as well as correlates of appearance-related constructs. Higher levels of perfectionism are associated with unfavourable social comparison evaluations [31,32], higher rates of upward counterfactual comparisons [33], and higher levels of body dissatisfaction [34]. This suggests that further research is necessary to explore relationships between perfectionist attitudes, unfavourable comparisons, and negative self-evaluations. Higher levels of narcissism are also associated with body dissatisfaction in extreme under- or overweight individuals [35], as well as increased social comparison orientation [36]. Taken together, we consider perfectionism and narcissism as potential moderating variables in the comparison process and assess their associations with aspects of appearance-based comparisons.

## The present research

This study set out to achieve two aims: first, to test the viability of a self-report measure of multi-standard comparisons related to self-evaluation of appearance and engendered affective reactions; second, to explore these evaluations and reactions in context of comparison correlates of wellbeing and personality as potential mediating and moderating factors. As part of our first aim, we developed the Comparison Standards Scale to investigate specific standards from *social*, *temporal*, *criteria-based*, *dimensional*, and *counterfactual* comparisons. The scale assesses each standard per direction (upward and downward) for recent frequency of engagement, degree of perceived discrepancy, and affective impact. We explore the factor structure of the scale based on frequency item ratings and assess descriptive and psychometric properties using mean scores for frequency and sum scores for discrepancy and affective impact per comparison direction. We also calculate mean comparison Intensity scores as a product of frequency and discrepancy, which were highly correlated (.92 and .90 for upward and downward comparisons, respectively). For our second aim, we investigate how comparison intensity and affective impact scores are associated with comparison correlates of wellbeing and personality. To this end, we measured individual differences in self-reported levels of depression, anxiety, stress, self-esteem, perfectionism, and narcissism.

Hypotheses were not pre-registered, yet we expected exploratory factor analyses of all frequency items to identify upward and downward comparison factors, while factor analyses on items for each type of comparison standard (i.e., social, temporal, etc.) would identify one factor. A confirmatory factor analysis was also used to explore a bifactor structure based on frequency items. Upward comparison standards were expected to be the most frequently

reported. Corresponding affective impact ratings were expected to be mostly negative for upward comparisons and mostly positive for downward comparisons. Exceptions were expected for future temporal comparisons and compensatory dimensional comparisons based on motivational significance (e.g., downward social and upward future temporal comparisons are both potentially favourable evaluations) [1]. The calculation of comparison intensity scores was adopted post-data collection, thus subsequent analyses were exploratory. We explore inter-comparison correlations of intensity scores between upward and downward comparisons. For convergent and divergent validation of our novel scale, we investigate associations of comparison intensity with measures of appearance schemas, and social comparison intensity with SCS scores. We also explore associations of wellbeing, self-concept and personality measures with comparison intensity and affective impact per direction.

Using moderation and mediation models we explore the mediating role of *Appearance Concern* and the moderating role of *Negativity* on the associations between intensity and affective impact. Appearance concern is a composite of appearance schemas and physical attractiveness self-concept, reflecting a preoccupation with the dimension of comparison. We assume that frequent engagement in comparison to different standards and/or perception of large discrepancies between the self and the standard, will lead to a stronger preoccupation with appearance [37], which in turn influences engendered affective reactions. We also assess how the relationship between intensity and appearance concern is moderated by personality factors (narcissism and perfectionism). Negativity is a composite of depression, stress, anxiety, and self-esteem, reflecting a general experiential psychological wellbeing. These constructs are known to influence the comparison standards sought [38], the frequency of comparisons, and the degree of affective consequences [2,3].

## Methods

All original material of the online survey and the anonymized data can be found in the OSF supplement at https://osf.io/q3scp/?view_only=f109758e54d444858ec22c5c6cf94a34. Reliability for independent variables was assessed using Cronbach's Alpha according to accepted guidelines [39].

### Participants

Three hundred and sixty-one participants accessed the study link, and 300 participants completed the full survey (71.3% women, $n = 214$) with a mean age of 25.6 years (SD = 9.94, Range = 18 to 64). Participants were recruited from the University of Münster, local community spaces (e.g., supermarkets), and via posts in social media groups, using an advert with information about the study, contact information and an online link. Participants were included if they were over 18 years old and were native speakers of German (97.3%) or spoke German fluently (2.7%). The sample was predominantly made up of students ($n = 238$, 79.3%). Participants received a small monetary compensation or partial course credit in return for completion of all measures. This study was approved by the ethics committee of University of Münster.

### Comparison Standards Scale for Appearance (CSS-A)

We developed the CSS-A to assess upward and downward appearance-based comparisons across multiple comparison standards. We included items to explore the degree to which individuals report engaging in upward and downward comparisons via social, temporal, criteria-based, dimensional, and counterfactual standards regarding the dimension of one's own appearance, as well as the degree of comparison discrepancy and the affective reaction. We

**Table 1. Comparison standards and item descriptions for the CSS-A.**

| Type | Standards (items) | Example of *upward/downward* directions |
|---|---|---|
| **Social** | Familiar (4 items: SC1, SC2, SC3, SC4) | Comparing with a close friend/family member that looks *better/worse* than you. |
| | Unfamiliar (4 items: SC5, SC6, SC7, SC8) | Comparing with a stranger/celebrity that looks *better/worse* than you. |
| **Temporal** | Past (2 items: TC1, TC2) | Thinking that you used to look *better/worse* than currently. |
| | Future[a] (2 items: TC3, TC4) | Thinking that you might look *better/worse* in the future than currently. |
| **Criteria-based** | Ideal/Feared (2 items: CBC1, CBC2) | Imagining *the best/worst* you could possibly look in relation to your current appearance. |
| | Ought (2 items: CBC3, CBC4) | Thinking about how people your age and gender should look, and that you look *worse/better* than this. |
| **Dimensional** | Compensatory[a] (2 items: DC1, DC2) | Thinking that *you have other personal attributes* that make up for what you lack in *appearance*. Thinking that *your appearance* makes up for what you lack in *other personal attributes*. |
| | Salience (2 items: DC3, DC4) | Thinking of your appearance as a uniquely *worse/better* attribute compared to your other personal attributes. |
| **Counterfactual** | Subtractive (2 items: CFC1, CFC2) | Thinking that if certain things had not happened in the past, your appearance would now be *better/worse* than currently. |
| | Additive (2 items: CFC3, CFC4) | Thinking that if certain things had happened in the past, your appearance would now be *better/worse* than currently. |

[a]We expected the valence of affective impact for these items to opposite compared to other items, i.e., upward–positive, downward–negative.

developed scale items in English and then translated them to German. We then refined the German items and back-translated them to English. Both native English and German speakers were involved in this process. This study used the German version and both English and German scales can be found in the OSF Materials supplementary folder under 'Comparison standards scales'.

Table 1 summarizes the types of comparison and formulations of the standards. For each standard (e.g., social familiar: a close friend), participants were asked to what extent they compared with the standard that looks better (upward) and looks worse (downward). Participants rated the degree to which they had engaged in each type of comparison in the past three weeks on six-point Likert scales (0 = *not at all* to 5 = *very often*). If participants indicated more than "0 –*not at all*", they were asked two follow up questions. First, they rated the degree of discrepancy with the standard by indicating how much better or worse they perceived the standard to be on a six-point Likert scale (0 = *not at all* to 5 = *much better/worse*). Second, respondents indicated the extent to which the comparison made them feel better or worse on a bipolar seven-point Likert scale for affective impact (-3 = *much worse* to +3 = *much better*). In total, the scale consists of 24 obligatory frequency items, of which 12 were upward and 12 were downward items, and an additional 48 potential sub-items addressing discrepancy and affect. We chose a three-week recall period based on feedback from a pilot test (*N* = 10; data can be found in the OSF supplement under 'Pilot data') and incorporated suggestions for phrasing and understanding into the final version. In the current sample Cronbach's alpha was good for the upward items (.80) and acceptable for the downward items (.71).

Mean scores of comparison frequency were calculated per direction (i.e., upward and downward), while sum scores were used for comparison discrepancy and affective impact per standard and direction, due to missing data by design (i.e., no response when a participant responded with 0 –*not at all* to the frequency item). The mean frequency scores reflect how often participants engaged with comparison standards, whereas discrepancy indicates the degree of perceived difference with the standard. Scores of comparison intensity were calculated by multiplying frequency and discrepancy ratings per item, with subsequent mean scores per comparison direction. The intensity variable represents the extent of the target vs standard evaluation outcome in the basic comparison process. Affective impact scores indicate the

engendered positive or negative affect participants felt from comparison evaluations. To assess the valence of each comparison standard, we ranked the mean scores for each affective impact item from most negative to most positive (see Figure in OSF supplementary figures). Psychometric properties are addressed in the results section.

## Measures of appearance and comparison

**Appearance Schemas Inventory-Revised (ASI-R).** The ASI-R assesses the cognitive-behavioural investment of individuals in their appearance across 20 items [40]. The inventory comprises two subscales, self-evaluative salience (twelve items, e.g. "My appearance is responsible for much of what's happened to me in my life") and motivational salience (eight items, e.g. "I spend little time on my physical appearance"). Each item is rated on a 5-point Likert scale (1 = *strongly disagree* to 5 = *strongly agree*). An average score of the items is used as an index of preoccupation with appearance, where higher scores indicate higher engagement with appearance schemas. We used total combined scores from the German version in this study, as the subscales in this version are reported to be valid only for specific samples such as students or individuals with body dysmorphic disorder [41]. Cronbach's alpha was very good (.87).

**Social Comparison Scale (SCS).** The SCS is a self-report scale of social rank. It consists of 11 items addressing various comparison dimensions on bipolar 10-point scales (e.g., 1 = *inferior* to 10 = *superior*, 1 = *unlikeable* to 10 = *likeable*). Participants rank themselves "in relation to others" [16] and scores are summed to yield an overall score between 11 and 110. Higher scores suggest a more positive self-evaluation compared to others, while lower scores suggest more negative self-evaluation. The SCS also provides three subscores of group fit, social rank, and attractiveness. The present study used the total score and the attractiveness subscale score from a German adaptation [42]. Cronbach's alpha was very good for the full scale (.89) and acceptable for the attractiveness subscale (.71).

## Wellbeing, self-concept, and personality measures

**Depression, Anxiety and Stress Scales (DASS-21).** The DASS-21 measures depression, anxiety and stress via three subscales [43,44]. This study used a German version of the DASS-21 [45] with Cronbach's alpha indicating excellent reliability for totals cores (.92), very good reliability for depression (.88) and stress (.89), and acceptable reliability for anxiety (.74). Each subscale consists of 7 items pertaining to the past week (e.g., "I felt that I had nothing to look forward to"), with participants using a 5-point Likert scale rating their agreement (0 = "*Did not apply to me at all*" to 4 = "*Applied to me very much, or most of the time*"). Scores were summed for subscales, as well as for a total score [45].

**Rosenberg Self-Esteem Scale (RSES).** The RSES is a well-established measure of general self-esteem consisting of ten items relating to positive and negative feelings about the self [46]. Participants rate their agreement on a 4-point Likert scale (0 = "*strongly disagree*" to 3 = "*strongly agree*"). This study used a German version [47] and Cronbach's alpha was very good (.89).

**Multidimensional Self-Concept Scale–Physical attractiveness subscale (MSCS-P).** The MSCS is a self-report measure of self-concept for the hierarchical facet model of self-esteem [48]. The current study used the German version [49] and included only the physical attractiveness subscale that consists of five items (e.g., "How confident are you that others see you as being physically appealing?"). Participants are asked to rate each item on a 7-point Likert scale (1 = "*not at all*" or "*never*" to 7 = "*very often*" or "*always*"). A sum score of the items is used as

an index of appearance-based self-esteem, where higher scores indicate higher self-esteem. Cronbach's alpha was acceptable in the current sample (.76).

### Narcissistic Admiration and Rivalry Questionnaire short scale (NARQ-S)

The NARQ-S is a short version of a narcissism scale that measures two related dimensions: admiration and rivalry [50]. Participants rate their agreement with statements on a 6-point Likert scale (1 = "*do not agree at all*" to 6 = "*agree completely*"). Statements are associated with the narcissism dimensions, which represent the intrapersonal strategies of self-promotion and self-defence that are used to maintain the grandiose self (e.g., "I want my rivals to fail"). Cronbach's alpha was acceptable for the admiration subscale (.76) and questionable for rivalry subscale (.68).

**Short Almost Perfect Scale (SAPS).**   The SAPS is a shortened version of the Almost Perfect Scale-Revised [51], consisting of eight items that measure two elements of perfectionism: high performance expectations (standards; e.g., "I set very high standards for myself") and self-critical attitudes associated with performance evaluation (discrepancy; e.g., "Doing my best never seems to be enough"). Participants rate their agreement with statements on a 7-point Likert scale (1 = "*strongly disagree*" to 7 = "*strongly agree*"), with higher scores indicating higher levels of perfectionism. Cronbach's alpha was good for the standards (.83) and discrepancy (.85) subscales.

### Statistical analysis

We had an A PRIORI assumption of two latent factors of the CSS-A based on upward and downward items. However, this is the first time the structure is investigated and there is no data available to predict if standard-based subscales (e.g., social, temporal) would also be found as latent factors. To explore the factor structure we first used exploratory factor analysis (EFA) with oblique rotation in IBM SPSS Statistics, assessing loadings of the frequency items for comparison direction and for each of the five comparison standards (i.e., social, temporal, criteria-based, dimensional, counterfactual) to see if the underlying latent factors could be statistically validated [52]. A confirmatory factor analysis (CFA) was also performed with a bifactor model using Lavaan [53] in RStudio to assess the structure of the scale in which all items load on a general factor that reflects a common construct (i.e., appearance comparisons), yet also specifies orthogonal factors that represent the unique variance of items that is not accounted for in the general factor, i.e., upward and downward comparisons (for more information see Chen et al. [54]). Although upward and downward comparisons are not mutually exclusive, we aimed at assessing their unique variance using the bifactor model. The common variance should be accounted for in the general factor (appearance comparison). The specific upward and downward factors should then represent a largely mutually exclusive variance that is not accounted for by a general factor, therefore we define two specific factors in our model that represent upward comparisons and downward comparisons (see Fig 1). As recommended for ordinal data, we used the weighted least squares mean and variance adjusted (WLSMV) estimator [54]. Comparative Fit Index (CFI) and Tucker–Lewis Index (TLI) values > 0.95 indicate good fit and values > 0.90 indicate acceptable fit, root mean square error of approximation (RMSEA) and Standardized Root Mean Square Residuals (SRMR) values < .05 indicate good fit and values < .08 indicate acceptable fit [55,56].

Further analyses explored linear associations between comparison intensity, affective reaction, and the ASI-R, SCS, DASS-21, RSES, MSCS-P, NARQ-S, and SAPS. This included mediation modelling with moderated direct effects (see Fig 2) to test whether the association between comparison intensity and affective impact are mediated by Appearance Concern and moderated by Negativity. Using the averaging method of related constructs [57], we calculated

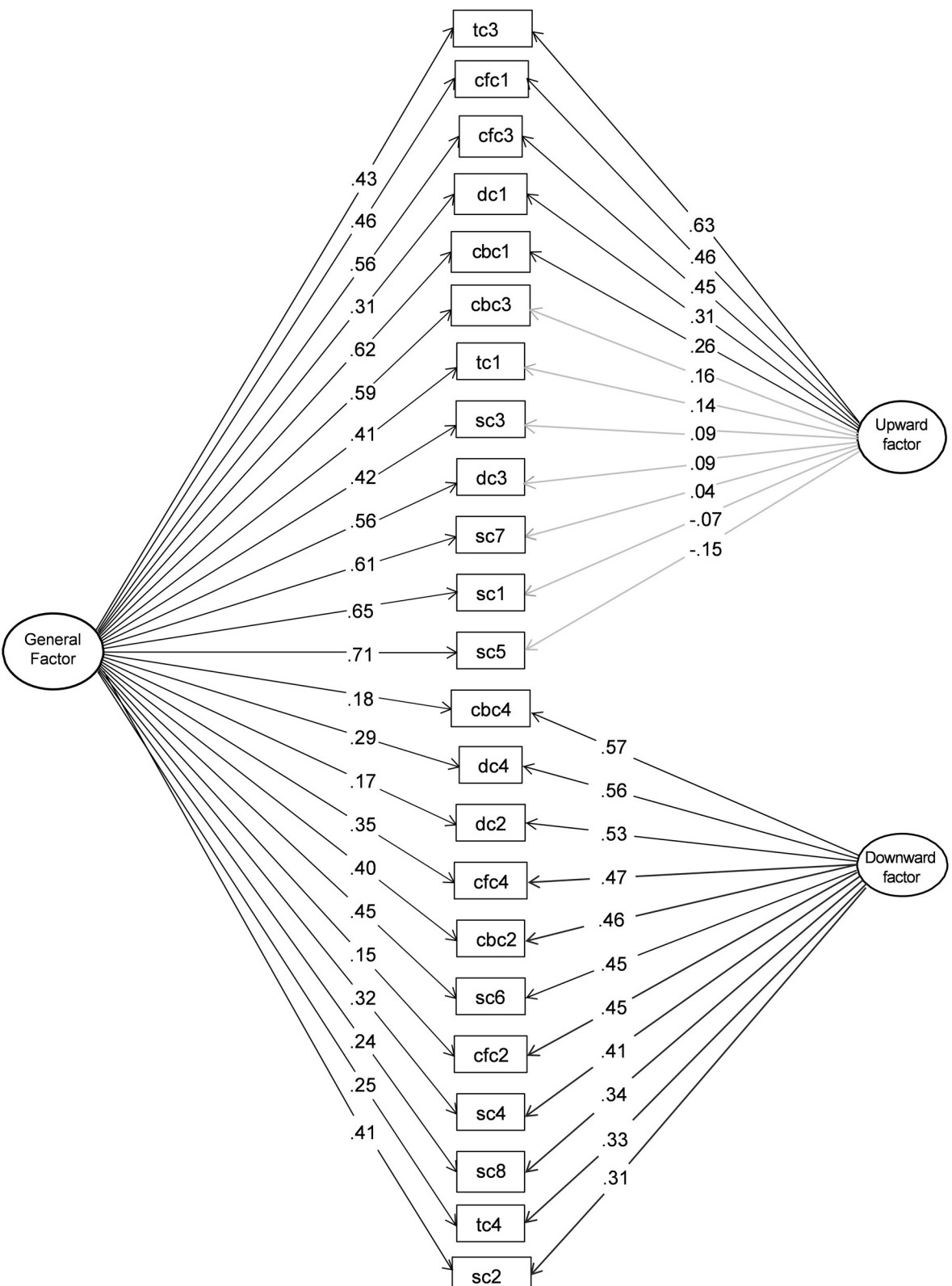

**Fig 1. Optimised Bifactor model of comparison direction for the Comparison Standard Scale.** Grey lines indicate non-significant loadings. Oddly numbered items address upward comparisons, and evenly numbered items address downward items; SC = Social comparison; TC = Temporal comparison; CBC = Criteria-based comparison; DC = Dimensional comparison; CFC = Counterfactual comparison.

a) Moderated mediation model for upward comparisons

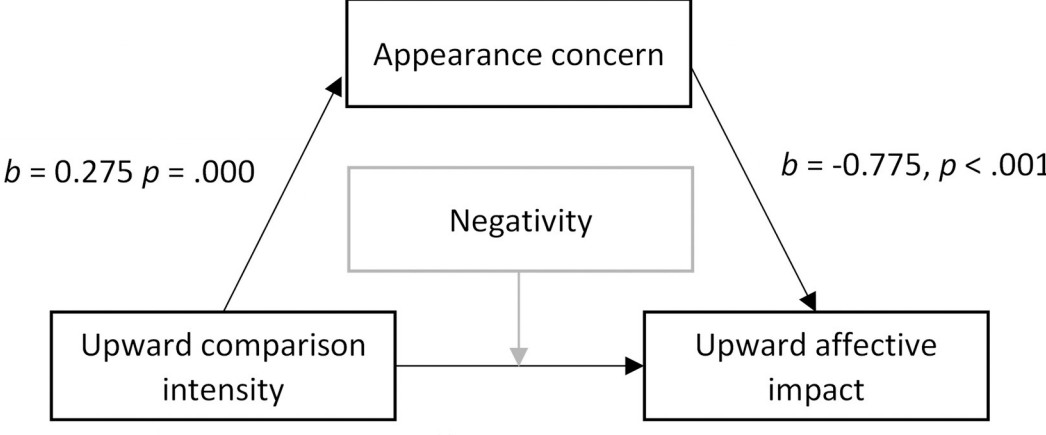

Direct effect, $b$ = -0.436, p < .001
Indirect effect, $b$ = -0.213, 95% CI [-0.319, -0.109]

b) Moderated mediation model for downward comparisons

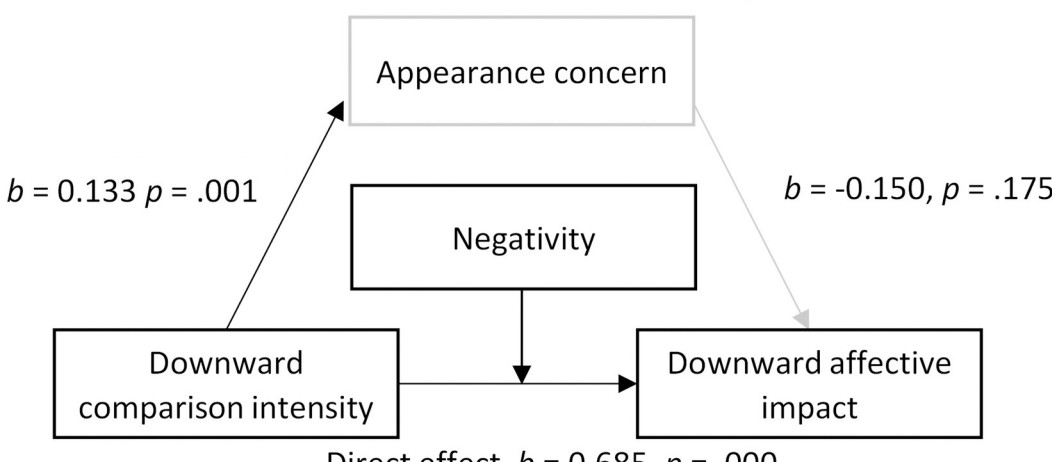

Direct effect, $b$ = 0.685, $p$ = .000
Indirect effect, $b$ = -0.020, 95% CI [-0.056, 0.009]

c) Personality moderated mediation model for upward and downward comparisons

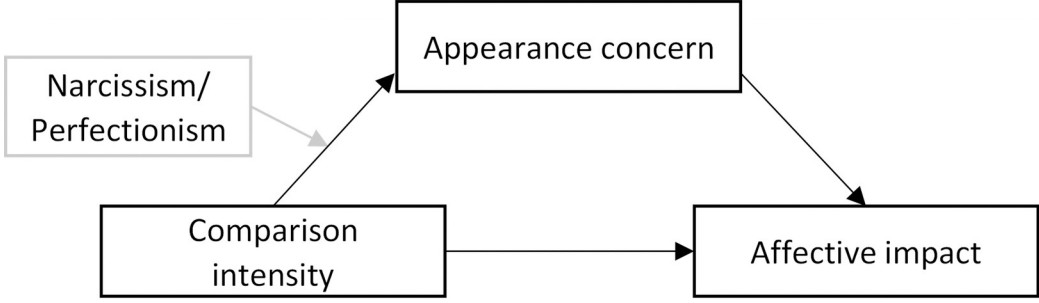

**Fig 2. Simple mediation models with moderated direct effects for appearance concern and negativity.** Grey lines represent non-significant effects.

the appearance concern variable by subtracting z-scores of the MSCS-P from those of the ASI-R, as negatively correlated measures ($r = -.324$, $p < .001$), and calculated the *Negativity* variable by subtracting z-scores of the RSES from the z-scores of the DASS-21 total, as negatively correlated measures ($r = -.640$, $p < .001$). The two composite variables Appearance Concern and Negativity correlated positively at a moderate level, $r = .504$, $p < .001$. In the case of significant mediation via appearance concern we further explored the personality effects of narcissism and perfectionism. To conduct moderation and mediation analyses we used conditional process modelling as described by Hayes [58] using the PROCESS macro in SPSS. Gender differences are assessed using independent t-tests; partial correlations between intensity, impact and independent variables can be found under the OSF supplementary gender analyses folder. The code to reproduce all analyses presented in this paper can be found in the OSF Materials supplement under 'Data and syntax' and additional results can be found under 'Supplementary results'.

Alpha was set at 0.05 (two-tailed) for all tests. The large number of individual tests increases the risk of alpha-error inflation; therefore, we used the Benjamini–Hochberg procedure to rank $p$-values using a false discovery rate of 0.10 [59]. Eight out of 71 significant alpha $p$-values fell below the critical value ($\alpha = 0.009$). Rankings can be found in the OSF supplement.

## Results

### Scale structure

Principal axis factor analyses with oblique rotation (direct oblimin) were performed on the frequency items of the CSS-A to evaluate item loading on factors per comparison direction and standard. An initial analysis of all 24 frequency items was run to examine the eigenvalues and the scree plot, which indicated only two reliable factors to retain. The analysis was run again with a fixed number of 2 factors to extract and the Kaiser–Meyer–Olkin (KMO) measure verified the sampling adequacy, KMO = .77. Table 2 shows the factor loadings after oblique rotation, where the items that cluster together load separately on two factors, with factor 1 representing upward comparisons and factor 2 downward comparisons. As a guide, we included items based on rotated factor loadings of >.30, despite some low communalities (< .30). Two items did not meet this criterion (TC2 and DC1). However, only for the Upward factor would removing an item (DC1) minorly improve Cronbach's alpha (.80 to .802). Hence, we included all items as these analyses were exploratory in nature and reflect appearance-based comparison habits. We anticipate that adapting the scale for other comparison dimensions (e.g., wellbeing or performance) will yield quite different loadings with the same items, thus recommend retaining/removing items based on Cronbach's alpha.

The factors explained 17.6% and 5.9% of the total variance respectively and analyses indicate acceptable reliability for both the upward factor ($\alpha = .79$) and downward factor ($\alpha = .71$). Factor analyses per comparison standard revealed only social comparison items provided a satisfactory KMO value above .60 and a distinct point of inflexion on the scree plots. Criteria-based comparisons provided a mediocre KMO value (.59) and an unclear inflexion on the scree plot. All other comparison standards (temporal, dimensional and counterfactual) provided poor KMO values and no clear inflexion on the scree plots. The results of the factor analyses for all items per direction and for items per each comparison standard can be found under 'Factor analyses' in the OSF supplementary results.

**Table 2. Comparison standards scale descriptives.**

| Item[a] | Standard | Frequency (S.E.) | n (>0) | Rotated factor loadings for frequency | |
|---------|----------|------------------|--------|---------|---------|
| | | | | **Factor 1** | **Factor 2** |
| SC1 | Close friend | 2.15 (.089) | 244 | **.585** | .009 |
| SC2 | | 1.41 (0.83) | 184 | .197 | **.351** |
| SC3 | Family | .67 (.074) | 82 | **.374** | -.015 |
| SC4 | | .41 (.053) | 63 | .064 | **.370** |
| SC5 | Acquaintance/ colleague | 1.71 (.088) | 203 | **.500** | .120 |
| SC6 | | 1.10 (.076) | 155 | .111 | **.513** |
| SC7 | Stranger/ celebrity | 2.31 (.104) | 226 | **.428** | .199 |
| SC8 | | .74 (.069) | 106 | -.008 | **.375** |
| TC1 | Past self | 1.59 (.096) | 183 | **.455** | -.085 |
| TC2 | | 1.69 (.095) | 196 | -.055 | **.271** |
| TC3 | Future self | 2.40 (.102) | 230 | **.615** | -.107 |
| TC4 | | 1.03 (.077) | 146 | .068 | **.328** |
| CBC1 | Ideal/feared self | 2.39 (.104) | 229 | **.545** | .152 |
| CBC2 | | .49 (.061) | 70 | .093 | **.468** |
| CBC3 | Ought self | 1.22 (.088) | 144 | **.615** | -.045 |
| CBC4 | | .97 (.079) | 129 | -.022 | **.472** |
| DC1 | Attribute compensation | 1.93 (.096) | 208 | **.251** | .137 |
| DC2 | | .72 (.069) | 105 | -.026 | **.454** |
| DC3 | Attribute salience | .33 (.049) | 53 | **.328** | .124 |
| DC4 | | .36 (.052) | 55 | .049 | **.470** |
| CFC1 | Should not have been | 1.33 (.097) | 149 | **.562** | -.058 |
| CFC2 | | .49 (.059) | 74 | -.102 | **.487** |
| CFC3 | Should have been | 1.08 (.087) | 128 | **.578** | .037 |
| CFC4 | | .32 (.052) | 45 | .007 | **.480** |

[a]All odd numbers are upward items, all even numbers are downward items.

Abbreviations. SC = social comparison; TC = temporal comparison; CBC = criteria-based comparison; DC = dimensional comparison; CFC = counterfactual comparison.

To provide further insight into the scale structure and how this may reflect the importance of specific comparisons in relation to appearance, we tested the two-factor (upward and downward comparisons) structure of the CSS-A in a confirmatory factor analysis using a bifactor model. The bifactor model allows us to test if there is unique variance in frequency items beyond a general appearance comparison factor due to upward and downward latent factors. We defined two specific factors of upward comparisons and downward comparisons and one general comparison factor in a bifactor model. The initial model including all items fit the data adequately $\chi^2(228)$ = 465.87, $p < .001$; RMSEA = 0.059 (90% CI: 0.051–0.067); SRMR = .084 and CFI = 0.939. However, one item (TC2, upward past temporal) was not a significant estimator of the general comparison factor and was removed to test an optimised version of the model. The optimised bifactor model (Fig 1) showed improved fit $\chi^2 (207)$ = 341.79, $p < .001$; RMSEA = 0.047, (90% CI: 0.038–0.055); SRMR = .078; CFI = 0.964. The estimated loadings for all items were significant for the general factor, however this was not the case for the upward factor. Only items TC3 (future temporal), CFC1 and CFC3 (counterfactuals), DC1 (compensatory dimensional), and CBC1 (ideal criteria-based) loaded significantly on the upward factor. The SC5 item (social acquaintance) negatively loaded on the upward factor, while having the strongest loading on the general factor, indicating that the general factor is defined by social

standards. The significant loadings of the other upward items on the general factor indicate that the model represents rather typical appearance comparison standards in the sample, where upward social items are predominant. Shared variance in the downward items is accounted for both in the general factor and the unique downward factor, indicating that downward standards occur in concert with upward comparisons, yet represent more unique appearance comparisons than upward standards.

Taken together, EFAs show that scale items adequately reflect upward and downward factors and that multiple types of standards contribute to these, yet social standards are most reliable as a comparison factor for appearances. The CFA supports this, where social standards are most relevant to the general factor. Variance in downward items is also accounted for in the general factor, indicating that it represents typical appearance comparisons. The unique variance accounted for in the upward and downward factors suggests that certain standards are less typical in appearance comparisons, especially downward standards.

### Properties of the CSS-A

**Descriptives of comparison frequency, discrepancy, intensity, and affective impact.**
Table 3 shows the means and standard errors for comparison frequency, discrepancy, intensity, and affective impact ratings per direction, as well as intra-standard correlations. All participants engaged with at least one comparison standard. Upward comparisons were reported by 99.7% of participants and downward comparisons by 96.0% of participants. The upward social comparison of a 'close friend' was the most frequently reported standard, while the downward 'additive' counterfactual comparison (i.e., if certain things had happened in the past) was the least reported standard. For affective impact, upward comparisons were characterized by mostly negative valence and downward comparisons by mostly positive valence as expected (temporal future items and dimensional compensatory items being the exceptions). However, the standards of downward feared (CBC2; $M = -0.63$, $SE = 0.17$) and downward attribute salience (DC4; $M = -0.16$, $SE = 0.15$) were unexpectedly negatively valenced. Nonetheless, means for downward salience items (DC2, DC4), and downward ideal (CBC1) were close to zero, indicating low to negligible affective impact. For individual items, the upward ought criteria-based standard (CBC3) had the most negative affective impact mean score of -1.12 from 48% of participants, while the downward ought criteria-based standard (CBC4) had the most positive mean of 1.04 from 43% of participants. Gender differences were found for upward comparisons, where women reported higher intensity scores, $t(194.13) = -3.91$, $p < .001$ ($M_{male} = 4.01$, $SD = 2.90$; $M_{female} = 5.60$, $SD = 3.74$), and more negative affective impact than men $t(179.59) = 3.93$ $p < .001$ ($M_{male} = -3.87$, $SD = 4.08$; $M_{female} = -6.35$, $SD = 5.03$).

### Associations between CSS-A and other measures of individual differences

Table 4 displays the correlation coefficients for associations of comparison intensity and affective impact with measures of appearance schemas, the social comparison scale, depression,

**Table 3. Means, standard errors, intra-standard correlations per comparison direction for frequency, discrepancy, intensity, and impact scores.**

|  | Combined scale Means (S.E.) | | | | Intra-standard correlations | | | |
|---|---|---|---|---|---|---|---|---|
|  | Frequency | Discrepancy | Intensity | Impact | Frequency ~ Discrepancy | Frequency ~ Impact | Discrepancy ~ Impact | Intensity ~ Impact |
| Upward | 1.59 (0.05) (N = 300) | 20.16 (0.60) (N = 299) | 5.13 (0.21) (N = 299) | -3.57 (0.30) (N = 299) | .922** (N = 299) | -.512** (N = 299) | -.544** (N = 299) | -.550** (N = 299) |
| Downward | 0.81 (0.03) (N = 300) | 10.92 (0.41) (N = 288) | 2.19 (0.12) (N = 288) | 2.01 (0.19) (N = 288) | .902** (N = 288) | .286** (N = 288) | .346** (N = 288) | .310** (N = 288) |

** p < .009.

**Table 4. Correlations per comparison direction and standard of CSS-A intensity and impact variables with independent and composite variables.**

| | Mean (SD)<br>N = 300 | Upward intensity<br>N = 300 | Down intensity<br>N = 288 | Upward impact<br>N = 300 | Down impact<br>N = 288 |
|---|---|---|---|---|---|
| **ASI-R** | 3.23 (0.58) | .432** | .265** | -.349** | -.021 |
| **SCS** | 66.95 (15.27) | -.152** | .033 | .277** | .216** |
| **DASS-D** | 4.66 (4.38) | .375** | .157** | -.346** | -.201** |
| **DASS-A** | 2.61 (3.15) | .237** | .185** | -.219** | -.094 |
| **DASS-S** | 6.16 (5.04) | .296** | .248** | -.284** | -.109 |
| **DASS total** | 13.43 (10.96) | .354** | .230** | -.331** | -.157** |
| **MSCS-P** | 20.81 (5.33) | -.579** | -.008 | .507** | .218** |
| **RSES** | 22.02 (6.16) | -.341** | -.142+ | .398** | .271** |
| **NARQ-A** | 2.42 (1.11) | .114+ | .230** | -.089 | .053 |
| **NARQ-R** | 2.10 (1.05) | .058 | .181** | .179** | .156 |
| **NARQ total** | 2.26 (0.93) | .099 | .239** | .056 | .125+ |
| **SAPS-S** | 22.17 (3.93) | .050 | .008 | -.132+ | -.065 |
| **SAPS-D** | 16.16 (5.54) | .170** | .075 | -.175** | -.120+ |
| **Appearance concern** | 0 (1.63) | .622** | .168** | -.527** | -.146+ |
| **Negativity** | 0 (1.81) | .384** | .205** | -.403** | -.236** |

ASI-R = Appearance schema inventory revised; MSCS-P = Multidimensional self-concept Physical subscale; SCS = Social comparison scale; DASS = Depression anxiety stress scale; RSES = Rosenberg self-esteem scale; NARQ = Narcissistic admiration and rivalry questionnaire; SAPS (-S & -D) = Short Almost Perfect Scale (-standards & -discrepancy). Appearance concern and Negativity mean scores are zero based on z-scores.

** p ≤ .009

+ p > .009, < .05.

anxiety, stress, appearance self-concept, self-esteem, narcissism, perfectionism, and composite scores of appearance concern and negativity. All reported coefficients are zero-order correlations.

**Convergent and discriminant measures.** *Appearance schemas.* We expected moderate correlations between the appearance schemas (ASI-R) and comparison intensity scores for construct validity. Women reported higher scores on the ASI-R, $t(296) = 3.91$, $p < .001$, $d = 0.50$ ($M_{male} = 3.03$, $SD = .60$; $M_{female} = 3.31$, $SD = .54$), however controlling for gender did not change the strength of correlations. For overall upward and downward comparison intensity, correlations with the ASI-R were moderate and low respectively ($r = .432$, $r = .265$, $ps < .001$). These associations suggest that upward appearance-based comparisons are most valid for appearance schemas. Upward, but not downward, affective impact ratings were significantly correlated with the ASI-R ($r = -.361$, $p < .001$), indicating that increasing appearance schemas are associated with more negative affect for upward comparisons.

*Social comparison.* We calculated social comparison intensity variables (Upward $M = 5.88$, $S.E. = 0.26$; Downward $M = 2.42$, $S.E. = 0.15$, $Ns = 300$) to assess validity. Correlations between social comparison evaluations (SCS) and social comparison intensity scores were expected to be low. Although they both measure social comparison evaluations, the CSS-A social subscales specify standards and take frequency into account. Correlations between the SCS and upward and downward social comparison intensity were low and non-significant, respectively ($r(300) = -.175$, $p = .002$; $r(300) = .023$, $p = .690$), which were non-significant when the SCS attractiveness subscale was used ($r(300) = -.132$, $p = .022$; $r(297) = .043$, $p = .495$). This suggests that appearance-based upward social comparison intensity has a small inverse relationship with social rank, indicating discriminant validity.

**Mediation and moderation models.** Mediation and moderation analyses were exploratory and assessed the influence of appearance, wellbeing, and personality variables on the relationship between comparison intensity and affective impact. The subscales for narcissism and perfectionism were assessed as moderators of path a (intensity–appearance) for the mediation models; however, no significant effects were found so we only report the results for appearance and wellbeing variables. Gender is included as a covariate for all models due to significant gender differences in upward comparisons and appearance variables. Results for all mediation and moderation analyses can be found under the OSF supplementary results.

We used appearance and wellbeing measures that showed significant correlations with upward and downward comparisons on intensity and affective impact scores to investigate potential mediating/moderating factors of the comparison process. We created composite scores of relevant comparison correlates reflecting *Appearance Concern*–i.e., the idiosyncratic salience of the appearance dimension and an unfavourable appearance self-concept–and *Negativity*–i.e., low mental wellbeing and a negative self-esteem, reflecting higher levels of neurotic traits. In line with a process-based approach of comparison, we tested how appearance concern mediated the relationship between comparison intensity and affective impact for upward and downward comparisons, as well as the role of negativity as a moderator of these relationships (Fig 2). We tested mediation with moderated direct effects (PROCESS model 5) [58]. Variables were mean centered and confidence intervals used 5000 bootstrapped samples for PROCESS modelling. We also used heteroscedastic-consistent inferences for standard errors to account for homoscedasticity.

For the model of upward comparison (Fig 2, model a) results show the effect of comparison intensity on affective impact was partially mediated by appearance concern, $b$ = -0.213, $b$se = .055 95%, CI [-0.319, -0.109]. As zero is not within the CI, this indicates a significant indirect effect of comparison intensity on affective impact [58]. There was no significant moderation of negativity on the direct effect of upward comparison intensity on affective impact, $F(1, 293)$ = 3.14, $p$ = .077, $\Delta R^2$ = .01. For downward comparison (Fig 2, model b) there was no significant mediation of appearance concern on the association between comparison intensity and affective impact, $b$ = -0.052, $b$se = .017, 95% CI [-0.095, -0.016]. However, the interaction between downward comparison intensity and negativity was a statistically significant predictor of affective impact, $F(1, 282)$ = 7.29, $\Delta R^2$ = .032, $b$ = -0.161, $b$se = .060, 95% CI [-0.279, -0.044]; $t(282)$ = -2.70, $p$ = .007. There was a significant positive relationship between downward intensity and affective impact at low, $b$ = 0.980, $b$se = .015, 95% CI [0.684,1.276]; t(282) = 6.52, $p$ = .000, and medium, $b$ = 0.685, $b$se = .011, 95% CI [0.468,0.903]; t(282) = 6.20, $p$ = .000, but not high levels of negativity, $b$ = 0.391, $b$se = .016, 95% CI [0.076,0.706]; t(282) = 2.44, $p$ =. 015. The simple slopes in Fig 3 show that for individuals reporting higher negativity ratings, the positive association between higher intensity and greater positive affective impact is weaker, whereas for lower negativity scores it is stronger.

## Discussion

This study presents a novel approach to assess between-person differences in aspects of the comparison process as they relate to appearance self-perception. Results suggest that the CSS-A reliably assesses individual differences across five types of comparison standards and engendered affective reaction. Participants reported more upward than downward comparisons and comparison intensity was significantly associated with affective impact. Appearance concern partially accounted for the aversive affective consequences of upward comparisons, while the positive association between downward intensity and affective impact was weaker for individuals who reported higher negativity ratings.

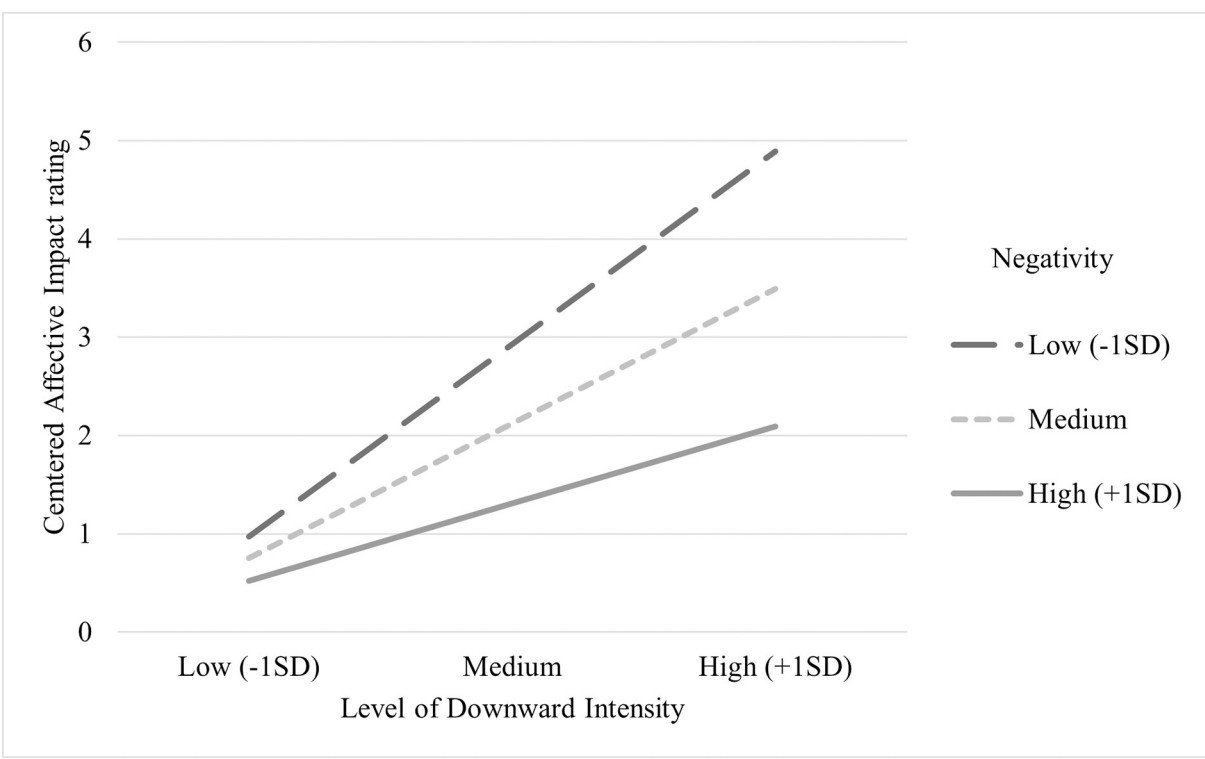

**Fig 3. Simple slopes for the regression of affective impact on intensity at three different levels of negativity for downward comparisons.**

### Structure & properties of the CSS-A items

Analyses of frequency items showed good reliability and the two-factor structure indicates that all types of standards are used in upward and downward appearance-based comparisons. Exploratory factor analyses for items per standard could only support extraction of a social standard factor, likely due to the greater potential of specific social standards (eight) compared to the other standards (four each). However, social standards are also more commonly used to inform appearance-based self-perceptions due to physical representations and socio-cultural phenomena such as advertising and social media, [e.g., 60–62]. This was supported by the CFA, where the general appearance factor was defined by a social item.

As expected, upward comparisons were more frequently reported than downward comparisons and were associated with predominantly negative affective impact, while downward comparisons were predominantly positive. In the EFA, the upward and downward comparison factors accounted for 25% of the common variance in frequency, however the downward comparison factor explained a relatively low amount of this, which usually indicates the items do not accurately measure a shared latent factor. Yet in the CFA, downward items accounted for variance in the general factor and the downward factor, where loadings were stronger for the latter. This indicates that while downward standards are utilised, they are less salient in appearance comparisons, supporting previous findings that they are not as common as upward standards [63,64]. Downward comparisons may be rather reactive, rather than self-initiated, and involve more flexible standards [65], thus being less consistent and more difficult to recall than upward comparisons.

Only future temporal, counterfactual, compensatory dimensional, and ideal criteria-based standards significantly contributed to a specific upward comparison factor, indicating that these captured less typical upward appearance comparisons. While taking the motivational

significance of comparisons processes into account [1], we anticipated the future temporal and compensatory dimensional standards as appetitive (i.e., favourable) upward comparisons. This was supported by the positive mean affective impact ratings for these items, while the counterfactual comparison items were aversive, indicating that the upward factor represents both appetitive and aversive standards outside of the typical appearance comparisons. Specific standards could prove useful for identifying atypical comparison habits that have (dys)functional properties. For example, according to the functional theory of counterfactual thinking [66], excessive upward counterfactual thinking is associated with higher negative affect and depressive symptoms, supported in a review of upward counterfactual thinking [28]. Further investigation of comparison standards as indicators of self-perception is necessary, particularly to establish what constitutes excessive or dysfunctional appearance comparisons, as well as if this varies between dimensions. Our results show that the CSS-A is a useful tool for future research to consider multiple comparison standards when investigating appearance self-evaluation processes, as well as providing a framework for research in other domains.

While appearance comparisons are largely shaped by social and upward standards, other comparison dimensions may be shaped by different types of standards. For example, comparisons involved in assessing one's wellbeing are predominantly based on upward comparisons using past temporal, social, and criteria-based standards, often with negative evaluations [3]. Whereas for academic and social performance, past and future temporal comparisons are most common and are associated with more positive self-evaluations than social and criteria-based comparisons [38]. We encourage further research using the CSS-A framework for other comparison dimensions to gain better insight into variations of the comparison process.

## The roles of *Appearance concern* and *negativity*

Both appearance concern and negativity showed stronger associations with upward comparison intensity and affective impact than downward comparison intensity and affective impact, and both composite variables were associated with negative affective impact. We tested a provisional process-based approach to assess the mediating and moderating properties of the composite variables on the relationship between comparison intensity and affective impact. For upward comparisons, higher comparison intensity with higher appearance concern was associated with negative affective impact. For downward comparisons, the association between comparison intensity and positive affective impact was moderated by negativity, where low negativity scores were indicative of higher positive affective impact as intensity scores were higher. These results support previous findings that dimension salience and psychological wellbeing influence comparison outcomes and subsequent affective reactions [2,3], and could explain findings such as why only some individuals experience the touted benefits of downward comparisons [67]. Future research should focus on these relationships in-situ due to the dynamic nature of cognitions, emotions, and behaviour.

While our conclusions may appear self-evident, very little research has investigated influences of the comparison process outside of self-esteem, group differences, or motives, especially beyond social comparisons. Although beyond the scope of this paper, perception of changeability of an attribute has also been identified as a moderator of self-evaluative process and outcomes [68], which can be applied to the comparison process and consequences, such as change in affect [1]. For example, if an individual makes an unfavourable comparison about their appearance and perceive this as a fixed entity, they are more likely to be threatened by the comparison and experience a negative change in affect, particularly if appearance concern is high. However, should the individual perceive appearance as malleable, this could lead to optimism and a positive change in affect.

## Measurement of the comparison process

Previous cross-sectional assessments of comparison standards have focused on unitary aspects of the comparison process, mostly the frequency of specific comparisons, such as in social [16] and counterfactual comparisons [69]. In a rare example of a multi-standard comparison scale, only frequency was assessed and the scale was limited to one item per direction and standard, and scoring ignored individual comparison standards [70]. Several reviews have reiterated that to fully understand the comparison process, it is important to consider various standards, direction of comparison, perceived (dis)similarity and engendered reactions [1,2,5]. The conceptualization of the CSS-A is therefore necessary to assess key aspects of the comparison process, with comparison intensity and affective impact variables providing respective indicators of comparison evaluation outcomes and engendered reaction. Literature investigating the associations of comparison habits with body-image perception and eating disorders have all but focused on social comparison [23,71,72], yet our results show that individuals engage with multiple types of standards. Observed gender differences in comparison intensity and affective impact ratings also occurred within upward standards, where women reported higher comparison intensity than men as well as lower negative affective impact scores. Investigating multiple comparison standards in body-image research could provide greater insight into individual differences when considering gender, as well as informing potential uses for clinical assessment and intervention.

## Strengths and limitations

Previous self-report measures have predominantly focused on singular types of comparison standards or have been limited to one aspect of the comparison process. We developed the CSS-A to measure multi-standard comparisons in context of perceived appearance. Our approach incorporates several theories of comparison standards that have found comparative thinking influences self-perception. Yet, the following limitations deserve to be mentioned. Despite recruiting a large sample and following the general rule of thumb of at least 10:1 ratio of participants to items for obligatory scale items [73], a larger sample and a better distribution of age and gender would have benefited validity and reliability. In addition, convergent validity regarding standards is limited given the lack of comparable measures and subsequent novel development of the scale. Thus, future studies with larger samples are required to further establish the characteristics of the CSS-A. Another potential limitation is that we focused on upward and downward comparisons, omitting the possibility of lateral comparisons. The general comparison processing model of self-perception by Morina [1] defines lateral comparisons as fulfilling an important role in the self-evaluation process. For example, lateral social comparisons tend to be reported just as often as upward social comparisons [5]; however, upward and downward comparisons across various standards have been reported more often than lateral comparisons for the dimension of wellbeing [3]. We did not explicitly refer to lateral comparisons, yet we assessed the degree of comparison discrepancy, where a value of zero may suggest a lateral comparison. Thus, using the intensity variables per direction, we attempt to control for lateral comparisons while assessing the typical outcome of upward or downward comparison evaluations.

## Future research

Our approach to assessing comparison as a process provides several avenues for future research. The current study focuses on appearance comparisons, however comparison tendencies represented by intensity and affective impact will likely differ depending on the comparison dimension, with contextual and individual differences influencing the comparison process

[1]. Therefore further research can adapt the CSS-A to other specific comparison dimensions, such as trauma-related counterfactual comparisons [29]. Using the process-based approach to assessment with experience sampling methods will also facilitate the assessment of multiple comparison dimensions, as well as providing within-person data to see if our findings occur at state-level. Previous experience sampling in diary studies have often focused on specific types of comparison standards [19–21], or were limited regarding information about comparison standards and self-evaluation [14]. Our approach could be applied to a diary method similar to Summerville and Roese [14] using prompts and a series of questions to assess what type of comparison standard was used, yet with a broader range of comparison standards, as well as additional questions addressing key aspects of comparison such as discrepancy. Finally, to examine the comparative impact of different types of comparison on appearance and engendered reactions experimental studies are required. More data on the comparison process at both trait and state levels will provide much needed insight on the aspects involved and how these contribute to beneficial and undesirable consequences.

## Conclusion

The abstract nature of comparison processes provides a fascinating challenge for understanding the dynamic construct of self-perception. This study proposes a new approach to defining and assessing aspects of the comparison process by using multiple standards. We introduce the CSS-A as a novel measure of multi-standard comparisons based on the dimension of appearance, which captures individual differences in habitual comparison tendencies. Our results indicate fundamental differences between upward and downward comparisons and underline why comparison should be treated as a process. Upward comparisons are very common, yet at the extreme spectrum are potentially dysfunctional, which coincides with excessive preoccupation of an attribute, in this case appearance. Downward appearance comparisons appear to be less common and subsequently less relevant to wellbeing and self-concept; however, the potential of downward comparisons for affect regulation may play a role in individuals with low psychological wellbeing. This paper provides a framework to be tested using different comparison standards. We hope that the CSS-A and our approach to assessment of comparison will encourage future research to consider the role of social, temporal, criteria-based, dimensional, and counterfactual comparison standards in relation to self-perception.

## Author Contributions

**Conceptualization:** Peter A. McCarthy, Thomas Meyer, Mitja D. Back.

**Data curation:** Peter A. McCarthy.

**Formal analysis:** Peter A. McCarthy, Mitja D. Back.

**Investigation:** Peter A. McCarthy, Nexhmedin Morina.

**Methodology:** Peter A. McCarthy, Thomas Meyer, Mitja D. Back, Nexhmedin Morina.

**Project administration:** Peter A. McCarthy, Nexhmedin Morina.

**Resources:** Peter A. McCarthy, Nexhmedin Morina.

**Software:** Peter A. McCarthy.

**Supervision:** Peter A. McCarthy, Thomas Meyer, Nexhmedin Morina.

**Validation:** Peter A. McCarthy.

**Visualization:** Peter A. McCarthy.

**Writing – original draft:** Peter A. McCarthy, Thomas Meyer, Mitja D. Back, Nexhmedin Morina.

**Writing – review & editing:** Peter A. McCarthy, Thomas Meyer, Mitja D. Back, Nexhmedin Morina.

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
