## [Decision Letter · Decision Letter 0]

25 May 2022

PONE-D-22-02594How we compare: a new approach to assess aspects of the comparison process for appearance-based standards and their associations with individual differences in wellbeing and personality measures.PLOS ONE

Dear Dr. Morina,

Thank you for submitting your manuscript to PLOS ONE. After careful consideration, we feel that it has merit but does not fully meet PLOS ONE’s publication criteria as it currently stands. Therefore, we invite you to submit a revised version of the manuscript that addresses the points raised during the review process.

Two expert Reviewers evaluated the manuscript and both gave encouraging opinions about the utility of the study. However, major revisions are needed. I suggest Authors to provide revisions taking into account both Reviewers' comments especially in terms of methodology/data analysis, clarity of background and possibly modification of specific terminology used in the article which I agree is not always clear. 

We look forward to receiving your revised manuscript.

Kind regards,

Stefano Triberti, Ph.D.

Academic Editor

PLOS ONE

**Journal requirements:**

Reviewers' comments:

Reviewer's Responses to Questions

**Comments to the Author**

1. Is the manuscript technically sound, and do the data support the conclusions?

Reviewer #1: Partly

Reviewer #2: Partly

2. Has the statistical analysis been performed appropriately and rigorously? 

Reviewer #1: Yes

Reviewer #2: Yes

3. Have the authors made all data underlying the findings in their manuscript fully available?

Reviewer #1: Yes

Reviewer #2: Yes

4. Is the manuscript presented in an intelligible fashion and written in standard English?

Reviewer #1: No

Reviewer #2: Yes

5. Review Comments to the Author

Reviewer #1: This is an thoughtful and interesting manuscript that introduces a new instrument to assess comparison behaviors, their association with well-being and attempts to disentangle the frequency, intensity and impact of upward and downward directional comparisons on appearance related constructs (like body self-esteem, depression, anxiety). The authors provide a compelling rationale for such an instrument that assesses multiple comparison domains- social, temporal, counterfactual. criterion-based, etc. This has been a serious gap so the instrument they have developed has considerable promise. However, the manuscript frequently used terms or descriptors that do not facilitate understanding. One such word is "engendered affect" -- in most cases, I don't understand whether the word engendered is needed. Their concept of intensity needs more justification as it is operationalized as a multiplicatively derived construct --comparison frequency and discrepancy seem to be the major components--but what if frequency is high, but upward discrepancy is low ----however, because discrepancy is low the subjects can assimilate their self-evaluations to the the better "other" -- why wouldn't subjects be overjoyed about that (I am realize I am exaggerating- using the word overjoyed). I may be missing something here --- maybe the statistical analyses adjust for this, but I can't tell in the current version of the paper.

Beyond these concerns, although the authors admit the sample mainly consists of college students (who may be at the stage, where appearance figures more importantly for than middle-aged or older adults, it is still a limitation Although I would allow to pass ---the research obviously involved considerable time and effort on the part of the authors.

I was surprised, however, to see the authors admitting the ambiguity with respect to causal direction, particularly when they often used the word "increased" or "decreased." The findings are complex and just how reverse causation produced the data patterns I cannot say. But the reader should receive a more upfront and focused warning.

I found the manuscript overly complex. I wondered whether the well-being measures might be reported elsewhere.

Several sentences are too lengthy and contain complicated phrasing. Sometimes I was just confused for examples:

line 174. line 176 what's the connection "ethnicity"? line 188, "congruent to personality type"- what does that mean?

line 204: stated as if unambiguously causal; line 222: "extent of upward and downward" -- unpack what that means?

Lines 230-235- is one long, complex, confusing sentence; break-up the content. Lines 293-294: "The items were "halved"- what does that mean and exactly why (I understand that number of soc comparison items are double the others, but doesn't halfing their values create a distorted index? Line 517: I could not understand the statement with the string of "and...and...and"'s. line 552, another lengthy, convoluted sentence.

Is there something worthwhile here? I believe there is, but certain words and phrases need to be replaced. The "intensity" operationalization needs more unpacking. The words "extent" and "engendered" create confusion.

And as noted earlier, I am not sure the personality and well-being results are warranted unless the rest of the paper can be made more understandable and coherent.

I would suggest giving the authors an opportunity to do that.

Reviewer #2: General comments:

First of all, I agree with the authors that the literature on comparison processes should be extended concerning different comparison standards, as proposed - hence, I find the aim of the paper/ research valuable in itself. The theoretical/ innovation aspect put aside, there are however several major methodological issues that warrant clarification.

Introduction:

Generally, the structure and content is adequate. However,

1. I wondered, why the INCOM was not used to validate the novel CSS-A?

2. I really do not see the rationale for why in the EFA the full range of frequency ratings was used, which was changed for the CFA to a dichotomous format. This questions the idea of CFA replicating the EFA findings; either the authors provide a better rationale for this, or the CFA should be rerun with the original scaling.

Methods:

3. For the SCS, (p. 14-15) only one reliability value is reported; however, two scores (total scale, attractiveness subscale) are later used. Please clarify.

4. Generally, the reporting of Cronbach's alpha is a bit weird; i.e., the authors write 'current sample indicated reliability with Cronbach's alpha ...' p. 15, l. 347. Please introduce a guideline for interpreting internal consistency and then report according to common standards; e.g. 'Cronbach's alpha for the scale was acceptable/ good/ very good in the present sample (.xx)'. The sentence in line 372 is incomplete (also concerns the reporting of internal consistencies). Please check carefully throughout.

Statistical Analyses/ Results:

5. Please also see my point of criticism no. 2. Why do you first run EFA (this is usually done to check how many latent factors, based on scree-plot) could be reasonably identified in the data; and the A PRIORI assume two factors for the CFA anyways? What's the point in doing EFA then in the first place? Please clarify.

6. The orthogonality assumption (line 381): is it reasonable to assume that for the two factors up- vs. downward comparisons? Are those really mutually exclusive?

7. Concerning the later conducted linear associations: please elaborate in how far the results of the CFA were taken into account; i.e., I do not think it is appropriate to use the different standards-subscales for correlation, for two reasons: A - number of items per subscale too low (< 3 items), B - CFA results contradicting computing sum scores/ average scores as proposed (i.e., some items not adequately loading on the two sub-factors).

Results

8. Please comment on the partly quite low communalities < .30 for some of the items (EFA); which cut-offs were used for communalities/ side-loadings and why?

9. The amount of explained variance for the second factor (5.9%) is relatively low; perhaps you could reflect on what this means concerning the relevance of this factor.

10. On p. 20 (ll. 443-448) you report goodness of fit-criteria; Cut-Offs for these should have been introduced in the Stat. Analysis section, including reference to the according guideline. In addition, you state that 3 items were removed, improving fit (which, descriptively, I agree). However, if I am not mistaken, the SRMR values are outside the scope of good fit (i.e., all above > .08)?

11. There is a large amount of individual tests in the following results sections (correlations, t-tests, and so on). Would it be perhaps reasonable, also to establish robustness of individual findings, to compensate for this huge amount of test and hence risk of alpha-error inflation, using e.g. FDR?`

12. The mediation and moderation analyses are, in my point of view, not well-founded in terms of theoretical basis for computing the Appearance Concern variable and Negativity variable (ll. 637 ff.).

Discussion

13. Please discuss whether the results might have looked differently using another comparison domain. In addition, what might be interesting is discussing incremental vs. entity theory as moderator (i.e., assuming stability vs. instability/ changeability of a certain attribute, might alter the impact of comparisons).

6. PLOS authors have the option to publish the peer review history of their article (what does this mean?). If published, this will include your full peer review and any attached files.

Reviewer #1: No

Reviewer #2: No

---

## [Author Response · Author response to Decision Letter 0]

12 Sep 2022

We appreciate the reviewers’ critical assessment of our paper and the chance to provide an improved and more understandable manuscript. We address the concerns raised point by point.

Reviewer #1: This is an thoughtful and interesting manuscript that introduces a new instrument to assess comparison behaviors, their association with well-being and attempts to disentangle the frequency, intensity and impact of upward and downward directional comparisons on appearance related constructs (like body self-esteem, depression, anxiety). The authors provide a compelling rationale for such an instrument that assesses multiple comparison domains- social, temporal, counterfactual. criterion-based, etc. This has been a serious gap so the instrument they have developed has considerable promise.

Our response:

We thank the reviewer for their kind words about our manuscript, as well as the subsequent feedback to help make the paper more understandable.

However, the manuscript frequently used terms or descriptors that do not facilitate understanding. One such word is "engendered affect" -- in most cases, I don't understand whether the word engendered is needed.

Our response:

We agree in some cases the word “engendered” was unnecessary and have deleted it when referring to the measure of affective impact to improve readability. In context of explaining reactions to comparison we retained “engendered” to emphasise these as consequences of the comparison process. 

Their concept of intensity needs more justification as it is operationalized as a multiplicatively derived construct --comparison frequency and discrepancy seem to be the major components--but what if frequency is high, but upward discrepancy is low ----however, because discrepancy is low the subjects can assimilate their self-evaluations to the the better "other" -- why wouldn't subjects be overjoyed about that (I am realize I am exaggerating- using the word overjoyed). I may be missing something here --- maybe the statistical analyses adjust for this, but I can't tell in the current version of the paper.

Our response:

We calculated the Intensity scores as a product of Frequency and Discrepancy (i.e., of these two variables only) because they were highly correlated: the correlation between Frequency and Discrepancy was .92 and .90 for upward and downward comparisons, respectively. We now clearly report the reason for combining the two (Lines 222-223). Furthermore, we have updated the definition of Intensity (Lines 310-313). We have also included the Frequency ~ Discrepancy correlations in Table 3. 

Beyond these concerns, although the authors admit the sample mainly consists of college students (who may be at the stage, where appearance figures more importantly for than middle-aged or older adults, it is still a limitation Although I would allow to pass ---the research obviously involved considerable time and effort on the part of the authors.

I was surprised, however, to see the authors admitting the ambiguity with respect to causal direction, particularly when they often used the word "increased" or "decreased." The findings are complex and just how reverse causation produced the data patterns I cannot say. But the reader should receive a more upfront and focused warning.

Our response:

We have replaced ambiguous references to “increased” with “higher”. In the entire paper there were no references to “decreased”, however we accept that this is written in context of the ambiguous “increased” phrases.

Regarding causal direction, we have deleted the ambiguous statement in Lines 745-747, given the evidence referenced in the introduction, such as Refs 1-3 in Line 57, and the diary studies in Lines 146 to 158.

I found the manuscript overly complex. I wondered whether the well-being measures might be reported elsewhere.

Our response:

All descriptives and correlations for wellbeing and personality measures are reported in Table 4. The text section has been removed. The number of correlations reported have also been reduced as standard subscales were removed in response to points raised by Reviewer #2.

Several sentences are too lengthy and contain complicated phrasing. Sometimes I was just confused for examples:

line 174. line 176 what's the connection "ethnicity"? line 188, "congruent to personality type"- what does that mean?

line 204: stated as if unambiguously causal; line 222: "extent of upward and downward" -- unpack what that means?

Lines 230-235- is one long, complex, confusing sentence; break-up the content. Lines 293-294: "The items were "halved"- what does that mean and exactly why (I understand that number of soc comparison items are double the others, but doesn't halfing their values create a distorted index? Line 517: I could not understand the statement with the string of "and...and...and"'s. line 552, another lengthy, convoluted sentence.

Our response:

Line 174 – 176: “DSM-5” was missing before “criteria”, reference to ethnicity in the research removed as out of context.

Line 188: changed to “comparing on specific dimensions that are congruent to depressive personality styles”, as described in the research article. 

Line 204: corrected to remove causal suggestion and new sentence started for readability, “This suggests that further research is necessary to explore relationships between perfectionist attitudes, unfavourable comparisons, and negative self-evaluations.”

Line 222: rewritten, “We also calculate mean comparison Intensity scores as a product of frequency and discrepancy, which were highly correlated.”

Line 231-235: broken into two sentences.

Line 293-294: standard subscales were removed in response to points by Reviewer #2.

Line 517: Line simplified for readability.

Line 552: broken into two sentences for readability.

Is there something worthwhile here? I believe there is, but certain words and phrases need to be replaced. The "intensity" operationalization needs more unpacking. The words "extent" and "engendered" create confusion.

Our response:

We have addressed “intensity” and the use of “engendered” in the previous comments. The use of the word “extent” in descriptions of variables has also been elaborated upon, providing more specific descriptions, for example in Line 310.

And as noted earlier, I am not sure the personality and well-being results are warranted unless the rest of the paper can be made more understandable and coherent.

I would suggest giving the authors an opportunity to do that.

Our response:

We anticipate that the adjustments to some of the definitions and the reduction of the subscales have made the paper more understandable and coherent. 

Reviewer #2: General comments:

First of all, I agree with the authors that the literature on comparison processes should be extended concerning different comparison standards, as proposed - hence, I find the aim of the paper/ research valuable in itself. The theoretical/ innovation aspect put aside, there are however several major methodological issues that warrant clarification.

Our response:

We thank the reviewer for their thorough feedback and input in considering our manuscript.

Introduction:

Generally, the structure and content is adequate. However,

1. I wondered, why the INCOM was not used to validate the novel CSS-A?

Our response:

For several reasons: The Social Comparison Scale (SCS) has been used more often to measure social comparison and meta-analyses have investigated associations between the SCS and psychopathology (Line 190). We could not justify two social comparison scales due to time restrictions. While the INCOM measures a tendency to engage in social comparisons cognitions and behaviours, we felt the SCS provided a more useful measure of self-evaluation relevant to comparison direction, as well as including an attractiveness subscale. However, we recognise that both scales have their limitations in respect to validation of the CSS-A.

2. I really do not see the rationale for why in the EFA the full range of frequency ratings was used, which was changed for the CFA to a dichotomous format. This questions the idea of CFA replicating the EFA findings; either the authors provide a better rationale for this, or the CFA should be rerun with the original scaling.

Our response:

We agree with the reviewer and have rerun the analysis with original scaling.

Methods:

3. For the SCS, (p. 14-15) only one reliability value is reported; however, two scores (total scale, attractiveness subscale) are later used. Please clarify.

Our response:

The reliability value has been added for the attractiveness subscale.

4. Generally, the reporting of Cronbach's alpha is a bit weird; i.e., the authors write 'current sample indicated reliability with Cronbach's alpha ...' p. 15, l. 347. Please introduce a guideline for interpreting internal consistency and then report according to common standards; e.g. 'Cronbach's alpha for the scale was acceptable/ good/ very good in the present sample (.xx)'. The sentence in line 372 is incomplete (also concerns the reporting of internal consistencies). Please check carefully throughout.

Our response:

The guideline from George & Mallery (2003) is introduced on line 259 and descriptions of internal consistency were updated accordingly.

Statistical Analyses/ Results:

5. Please also see my point of criticism no. 2. Why do you first run EFA (this is usually done to check how many latent factors, based on scree-plot) could be reasonably identified in the data; and the A PRIORI assume two factors for the CFA anyways? What's the point in doing EFA then in the first place? Please clarify.

Our response:

The A PRIORI assumption of two factors is based on the theory for designing the upward and downward items. However, this is the first time the structure is investigated and there is no data available to predict if standard-based subscales (e.g., social, temporal) would also be found as latent factors. Therefore, we did EFA first with each group of standard-based items, as well as all items together, to see if the underlying latent factors could be statistically validated (Schumacker & Lomax, 2010). The CFA was conducted to test the structure in bifactor model theory, which is addressed in point 6 below.

We have added this explanation in lines 379-386.

6. The orthogonality assumption (line 381): is it reasonable to assume that for the two factors up- vs. downward comparisons? Are those really mutually exclusive?

Our response:

The EFA used oblique rotation (line 380) as we assumed the two factors are not exclusive. However, we use the bifactor model specifically to assess their unique variance. The common variance should be accounted for in the general factor (appearance comparison). The specific upward and downward factors should then represent a largely mutually exclusive variance that is not accounted for by a general factor.

We have included this explanation in lines 391-396.

7. Concerning the later conducted linear associations: please elaborate in how far the results of the CFA were taken into account; i.e., I do not think it is appropriate to use the different standards-subscales for correlation, for two reasons: A - number of items per subscale too low (< 3 items), B - CFA results contradicting computing sum scores/ average scores as proposed (i.e., some items not adequately loading on the two sub-factors).

Our response:

We agree that the statistical justification for the standards-subscales is lacking and have removed these analyses for the linear relationships.

The EFA supports computing of the two subscales. The CFA was used to assess a general factor and the results rather represent typical comparison standards, defined by upward social standards. We have now elaborated on this in lines 477-488 and discussed the implications for this in relevant parts of the discussion, e.g., lines 621 & 637 onwards.

Results

8. Please comment on the partly quite low communalities <.30 for some of the items (EFA); which cut-offs were used for communalities/ side-loadings and why?

Our response:

As a guide, we included items based on rotated factor loadings of >.30, despite some low communalities (<.30). Two items did not meet this criteria (TC2 and DC1). However, only for the Upward factor would removing an item (DC1) minorly improve Cronbach’s alpha (.80 to .802). Hence, we included all items as these analyses were exploratory in nature and reflect appearance-based comparison habits. We anticipate that adapting the scale for other comparison dimensions (e.g., wellbeing or success) will yield quite different loadings with the same items, thus recommend retaining/removing items based on Cronbach's alpha.

We include this explanation in lines 443-449.

9. The amount of explained variance for the second factor (5.9%) is relatively low; perhaps you could reflect on what this means concerning the relevance of this factor.

Our response:

We reflect on this in the discussion, lines 627-636.

10. On p. 20 (ll. 443-448) you report goodness of fit-criteria; Cut-Offs for these should have been introduced in the Stat. Analysis section, including reference to the according guideline. In addition, you state that 3 items were removed, improving fit (which, descriptively, I agree). However, if I am not mistaken, the SRMR values are outside the scope of good fit (i.e., all above > .08)?

Our response:

We introduced the cut-offs in the methods section, lines 396-401.

With the new CFA results, the SRMR is now <.08 after removing only 1 item. However, as per our response to Point 8, we have included all items in further analyses of direction-based comparisons.

11. There is a large amount of individual tests in the following results sections (correlations, t-tests, and so on). Would it be perhaps reasonable, also to establish robustness of individual findings, to compensate for this huge amount of test and hence risk of alpha-error inflation, using e.g. FDR?

Our response:

We have used the Benjamini–Hochberg procedure to rank p-values using an FDR of 0.1 and have detail this in lines 420-424. 

12. The mediation and moderation analyses are, in my point of view, not well-founded in terms of theoretical basis for computing the Appearance Concern variable and Negativity variable (ll. 637 ff.).

Our response:

Our theoretical basis for computing the composite variables has been elaborated on in lines 244-255, and explanation of the computation has been moved from the results to the methods-statistical analysis section.

Discussion

13. Please discuss whether the results might have looked differently using another comparison domain. In addition, what might be interesting is discussing incremental vs. entity theory as moderator (i.e., assuming stability vs. instability/ changeability of a certain attribute, might alter the impact of comparisons).

Our response:

We discuss how different comparison domains are likely to show different comparison patterns in lines 655-663

We also introduce perceived changeability of comparison dimensions in the discussion of moderators, and how this may be observed in appearance comparisons; lines 679-688.

---

## [Decision Letter · Decision Letter 1]

20 Dec 2022

How we compare: a new approach to assess aspects of the comparison process for appearance-based standards and their associations with individual differences in wellbeing and personality measures.

PONE-D-22-02594R1

Dear Dr. Morina,

We’re pleased to inform you that your manuscript has been judged scientifically suitable for publication and will be formally accepted for publication once it meets all outstanding technical requirements.

Kind regards,

Stefano Triberti, Ph.D.

Academic Editor

PLOS ONE

Additional Editor Comments (optional):

Reviewers' comments:

Reviewer's Responses to Questions

**Comments to the Author**

1. If the authors have adequately addressed your comments raised in a previous round of review and you feel that this manuscript is now acceptable for publication, you may indicate that here to bypass the “Comments to the Author” section, enter your conflict of interest statement in the “Confidential to Editor” section, and submit your "Accept" recommendation.

Reviewer #3: All comments have been addressed

2. Is the manuscript technically sound, and do the data support the conclusions?

Reviewer #3: Yes

3. Has the statistical analysis been performed appropriately and rigorously? 

Reviewer #3: Yes

4. Have the authors made all data underlying the findings in their manuscript fully available?

Reviewer #3: Yes

5. Is the manuscript presented in an intelligible fashion and written in standard English?

Reviewer #3: Yes

6. Review Comments to the Author

Reviewer #3: The article introduce a novel approach to assess habitual comparison processes.

I think all the reviewers' comment were addressed by the authors. So, I think that the manuscript can be considered for publication.

7. PLOS authors have the option to publish the peer review history of their article (what does this mean?). If published, this will include your full peer review and any attached files.

Reviewer #3: No

---

## [Editor Report · Acceptance letter]

3 Jan 2023

PONE-D-22-02594R1 

How we compare: a new approach to assess aspects of the comparison process for appearance-based standards and their associations with individual differences in wellbeing and personality measures. 

Dear Dr. Morina:

I'm pleased to inform you that your manuscript has been deemed suitable for publication in PLOS ONE. Congratulations! Your manuscript is now with our production department. 

Kind regards, 

on behalf of

Dr. Stefano Triberti 

Academic Editor

PLOS ONE